# An Exploratory Case Study of Mental Toughness Variability and Potential Influencers over 30 Days

**DOI:** 10.3390/sports7070156

**Published:** 2019-06-27

**Authors:** Ken Bradford Cooper, Mark Wilson, Martin Ian Jones

**Affiliations:** Department of Sport and Health Sciences, University of Exeter, St. Luke’s Campus Exeter, Exeter EX1 2LU, UK

**Keywords:** master athletes, running, within-person variability, qualitative

## Abstract

The purpose of this study was to explore whether mental toughness varies across a 30-day training block and whether such variability is associated with specific antecedents. This exploratory case study research investigated mental toughness variability using the Mental Toughness Index (MTI) with thirteen elite master runners across a series of self-selected training sessions, followed by interviews and follow-up questionnaires, to identify primary influencers of variability. There were significant differences in the MTI scores between baseline (before the training period), and the minimum and the maximum reported score over five self-selected training sessions (*p*’s < 0.004). The proceeding follow-up interviews and questionnaires then provided insights into factors influencing this intra-individual variability. These higher-level themes included foundational wellbeing, specific preparation, and actions utilized in the moment. This study is the first to demonstrate within-person MTI variability across specific training sessions and provides initial insights for both athletes and practitioners into potential influencers of mental toughness.

## 1. Introduction

“In every race, something within each athlete poses a simple question: ‘How bad do you want it?’ To realize your potential as an athlete, you must respond with some version of this answer: More. And then you have to prove it” [1] (p. 15).

How much drive and effort individuals put down in pursuit of their goals, how they control their attention and emotions, remain optimistic and hopeful, and resist the urge to lay down in the face of obstacles can influence what Fitzgerald and Marcora [1] refer to as athletic potential. Mental toughness, “a state-like psychological resource that is purposeful, flexible, and efficient in nature for the enactment and maintenance of goal-directed pursuits” [2] (p. 18), is an umbrella term that encompasses these positive psychological constructs. Umbrella constructs are useful in performance psychology because recent evidence indicates that participants do not make subtle conceptual distinctions between unique psychological dimensions when interpreting their experiences [2]. Therefore, one way to ‘prove it’ in the context of the opening quote, is to accumulate and integrate a range of adaptive thoughts, emotions, and behaviors; or in other words, to enhance and utilize your access to your reserves of mental toughness. 

In order to examine antecedents of mental toughness variability, it is important that the state-like nature of the construct first be evidenced [3,4]. For example, notable variability in mental toughness has been found to be present in elite youth tennis players across various situations [5]. These tennis players identified various components ranging from emotion to specific behaviors when perceiving a sense of mental toughness (vs. mental weakness) and indicated they experienced both mentally tough and mentally weak responses in their training and competition. Gucciardi, Hanton, Gordon, Mallett, & Temby [6] also found that within-person differences explained 56% of the mental toughness variance in a sample of university students, with the remaining 44% due to between-person differences. Finally, in a recent autoethnographic study, Cooper, Wilson and Jones [7] suggested that mental toughness varied across states within a series of ultra-endurance events, and also reflected upon potential antecedents (i.e., influencers) to the variation in this state-like nature of mental toughness. Identified influencers included emotional responses (anger, love), competition, encouragement, and a sense of ‘last chance’ opportunity.

In closing, Gucciardi [8] encouraged researchers to consider this within-person aspect of mental toughness rather than treating it as something that was a predetermined trait. Doing so would expand the opportunity for scholars, practitioners, and individuals across various settings to enhance the potential for goal achievement. To this end, this exploratory case study research was designed to examine variation in mental toughness over a training period and to ask the participants to consider the antecedents of any reported change in mental toughness. The first hypothesis was that significant within-person variability in mental toughness would exist over the 30-day study period. Following the quantitative data analysis, we explored the athletes’ perceptions of the antecedents of this perceived variability to try to develop a greater in-depth understanding of variation in mental toughness from the perspective of the athletes themselves.

## 2. Methods

### 2.1. Exploratory Case Study Design

The methodology selected for this study was that of an exploratory case study with a combination of written questionnaires and extensive oral open-ended semi-structured interviews. Case studies are utilized to “investigate contemporary phenomena in their natural context” [9] (pp. 30). Rather than setting up a laboratory environment, phenomena are studied in their chosen surroundings and pursuits to provide a more accurate context. More specifically, we elected an exploratory case study to look at, for the first time, elite master athletes and their level of mental toughness variability and, if identified as variable, the potential influencers of that variability. This design allowed participants to express their judgments, experiences and personal strategies without limitations and allowed the researchers to garner insight into real-life experiences with mental toughness. The exploratory case study is designed to investigate phenomena that allows formulated hypotheses to be tested where preliminary research is limited or lacking [10]. The exploratory case study provides “a broad discussion approach that enhanced the researchers’ learning from participants through exploration to fill in literature gaps” [11] (p. 151).

We determined to provide a thematic description of our entire data set to identify important themes for the reader [12] and integrated multiple steps to enhance the level of trustworthiness. These included credibility via member check, prolonged engagement and triangulation (data, investigator and method) as well as transferability via thick description and an audit trail for the purposes of dependability and confirmability [13], as noted in our Procedures section.

### 2.2. The Case

Following ethical approval from the authors’ university research ethics board, we recruited thirteen high-level master runners (see Table 1) to serve as participants. The selection of this population for this exploratory case study was made for several key reasons. First, as elite master athletes, they would be familiar with the need for (and the ability to reflect upon) the value of mental toughness in their athletic pursuits. Second, their lives would more closely represent the lives of the general population (e.g., careers, children, family, financial stressors) compared with many professional athletes. Lastly, they would have regular training sessions planned that would allow for reflection on mental toughness utilization (or lack thereof).

Acceptance was based on criteria of being high-level amateur athletes between 40 and 60 years of age who conducted high intensity training sessions and had completed a race time for an established distance event between one and 26.2 miles over the past 12 months placing them at 75% (upper level of Regional Class) or higher age-graded performance. Participants (eight men and five women; mean ± SD, age = 48 ± 5.4 years, age-graded performance = 81 ± 6.3%) first referenced a qualifying race score based on (http://www.mastersathletics.net/index.php?id=2595), to assess whether they met the inclusion criteria for the study. Potential participants then completed an initial screening questionnaire including the Holmes-Rahe Life Stress Inventory [14]. We required a score of fewer than 150 points in this inventory to help limit additional stress-related variables, based on an indication that ≤150 suggests a lower probability of developing a disorder related to stress [14]. Finally, participants identified five specific, self-selected high-RPE training sessions planned for the month of December and completed a baseline Mental Toughness Inventory [6].

### 2.3. Procedure (During 30 Days)

Following the initial selection process and baseline inventory, the participants then received a daily email directing them to a link where they completed a brief (<5 min) survey about various aspects from their life over the past 24 h and a second link within the same email to be used on the days of their five high-RPE sessions. The purpose of the daily survey was to stimulate reflection about a range of potentially salient influencing factors [15] and to act as an aide memoir for future data-prompted interviews [16] while the 30-day period provided a broader sampling within the life of each individual participant.

The factors tracked included sleep [17,18], caffeine intake [19], nutrition [20], stress [21], training fatigue [22], connection with significant other [23] and current injury or illness status [8]. Specifically, participants were asked to report on sleep quantity (in hours), sleep quality from 1 (poor) to 10 (excellent), nutrition from 1 (poor) to 3 (excellent), stress from 1 (low) to 5 (high), training fatigue from 1 (low) to 3 (high), connection with significant other from 1 (poor) to 5 (excellent), caffeine intake (in mg), and current injury/illness status from 1 (none) to 3 (affecting training). Not all these data were analyzed statistically in the current study, but all raw data are available in Appendix A.

We asked the participants to self-select their five planned high RPE sessions to allow them to continue their current training regimen, as well as to provide for the naturalistic approach, which studies the phenomena in its natural context [24]. On the five days of their self-selected, high RPE sessions within the 30 days, we invited the participants to complete a pre- and post-session Mental Toughness Index (MTI). 

#### Mental Toughness Index (MTI)

The MTI is an eight-item measure of mental toughness scored using the sum of items of a 7-point Likert scale from 1 (False, 100% of the time) to 7 (True, 100% of the time). We modified the wording of the original eight items [6] to match the context for the athletes without affecting the outcome of the assessment. For example, question one of the MTI states “I believe in my ability to achieve my goals” as a general question. For this study, it read “I believe in my ability to achieve my goals in this session.” Previous studies examining MTI internal reliability demonstrated both a high Cronbach’s α (0.900) and composite reliability (0.906) levels [25].

### 2.4. Procedure (Post 30 Days)

Following the completion of the 30-day baseline assessment, athletes were divided into two groups based on a combination of mental toughness variability and consistency of data provided. The division into two groups was to gather additional qualitative data related to the antecedents of variability in mental toughness noted during the initial data collection period in those showing the most notable within-person variability. Having demonstrated within-person variability in mental toughness from low to high days across the group of participants, we then sought to conduct exploratory research to consider whether participants identified antecedents of changes in state-like mental toughness. We adopted an exploratory and descriptive methodological framework, underscoring the epistemological position of qualitative research, meaning the real world does not exist independent of our knowledge [26]. This framework was utilized to guide this phase of data collection because exploratory research offers “new ways of seeing and perceiving how this segment of reality works, how it is organized, or, more specifically how and in what way different factors relate to each other causally” [27] (p. 139). The emphasis of exploratory research is on the causal mechanisms resulting in social phenomena, which was our primary purpose in this study. We adopted a qualitative description [28] methodology to direct sampling, data collection techniques, and data analysis decisions because it provided an effective means of garnering additional data for review and analysis.

Regarding the interview data, qualitative subjectivism was utilized. In line with the nature of qualitative description, we adopted elements of a grounded theory approach (e.g., open and axial coding, memoing, diagramming, constant comparison). This allowed the concepts to be derived from the data collected and open coded during this portion of the study [29,30] across two separate groups of participants.

#### 2.4.1. Group One Interviews

Five of the 13 athletes (two females and three males) were purposely selected for more detailed follow-up interviews based on the stated inclusion criteria of higher variability in MTI scoring and ongoing consistency of data provided. These five athletes and their maximum variation in MTI included (participant initials) TS (27), JB (20), RW (18), MR (16) and DM (16: see Table 2). The remaining eight athletes reported variation in MTI of six (RS, WM, SS and NO), 14 (PB), 15 (TR), 16 (GR) and 22 (EK) but demonstrated either less MTI variability or consistency in their data reporting. The interviews focused on participants’ mental toughness variability insights and their experiences during the study participation. We developed the interview questions based on concepts from prior mental toughness literature and the daily survey data that the participants collected over the 30-day training period from phase one (e.g., “What is mental toughness to you and please share thoughts about whether you see it as helpful in maximizing your performance”). Finally, the authors discussed the interview guide and challenged one another on the appropriateness of questions to develop the final guide.

The first author conducted the interviews by telephone because of the geographical diversity of the participants. While telephone interviews do not offer the potential for the identification of non-verbal or visual cues, the documented evidence of reduced effectiveness is lacking [28]. On the positive side, this method of interviewing does offer the advantage of convenience on the part of each participant and the absence of geographical limitations. The interviews averaged 41 ± 4.9 min in length and comprised the following topics: (a) role of mental toughness, (b) thoughts about the mental toughness variation they experienced and potential causes, (c) steps they take to increase their mental toughness, (d) thoughts about session-specific variation in mental toughness, and (e) potential application of mental toughness outside of athletics. Throughout the interview, the interviewer performed additional probing or clarification of responses [31].

Following the initial review of the transcribed interviews, a set of six clarification follow-up questions were then sent to each of these five interviewees to incorporate into the qualitative analysis for additional insights. These questions were: (1) What does being mentally tough mean to you? (2) Describe a time when you demonstrated mental toughness (3) Describe a time when you were not mentally tough (4) What do you believe are the top two elements that fuel or increase your mental toughness for/during a tough session or event? (5) What would you identify as your mental toughness “kryptonite”? Identify 1–2 elements that negatively influence your mental toughness for/during a tough session or event (6) Why do you care personally about influencing your mental toughness? 

We achieved data saturation to the point at which no additional data was being collected through further analysis. This was initially completed through the process of the detailed initial qualitative interviews, which sought to determine the influencers of the mental toughness variability. Saturation was then confirmed further through the inclusion of qualitative data garnered through the additional brief follow-up surveys with the other participants as noted below. The feedback provided by these additional athletes helped confirm this data saturation [32] on the insights collected about mental toughness variability from the interviews.

#### 2.4.2. Group Two Questionnaires

The other eight athletes from the initial population were comprised of those with less consistent daily tracking data provided (i.e., missing data) and lower maximum high to low MTI variability. These eight Group Two athletes completed a brief, five-item email follow-up questionnaire: (1) How do you define mental toughness? (2) What external factors in your life (things outside of your control) influence your mental toughness positively or negatively? (3) What internal factors (choices/decisions/plans) influence your mental toughness? (4) When you sense your mental toughness is lower than desired, are there things you can do to improve it? (5) Have you seen times in your life when your mental toughness seemed lower and if so, can you point to things you were doing differently or choices you were making that may have led to that lower mental toughness? Comments and descriptions were similar or supported those provided by the Group One athletes.

### 2.5. Data Analysis

At the completion of the 30-day survey period, self-reported data for each participant was collated from the online database. We first assessed variability in MTI scores by running a repeated measures ANOVA on the highest and lowest score from the five high-RPE training sessions and the baseline value from the beginning of the 30 days. Follow-up comparisons were run using Bonferroni corrected *t*-tests and effect sizes are reported using partial eta squared. Analyses were run on Jamovi (Version 0.9.1.7, Retrieved from https://www.jamovi.org). We also computed a coefficient of variation (mean/SD, expressed as a percentage) for the most pertinent survey data (i.e., those providing consistent responses); sleep quality, sleep quantity, stress, and significant other connection, for illustrative purposes.

All interviews were recorded and transcribed verbatim, resulting in a total of 51 single-spaced pages from the five Group One athletes. An additional five pages of unformatted email responses from the other eight (Group Two) athletes were compiled and spreadsheets summarizing low MTI comments, MTI patterns and answers to brief follow-up questions were gathered from all participants. With a focus on achieving qualitative description, the interview transcriptions and questionnaire response documents were repeatedly reviewed, analyzed and coded to identify themes and critical insights from the participating athletes about potential influencers of mental toughness. The goal was to “seek to discover and understand a phenomenon, a process, or the perspectives and worldviews of the people involved” [24] (p. 1) Responses to questions were coded, categorized and then developed into descriptive concepts. We moved beyond the participants’ literal descriptions in an attempt to interpret the data within the context of those literal descriptions [24], and raw data themes, first order grouping, and higher order themes were identified as shown in Figure 1. Raw data themes were initially developed by the primary author and then critically reviewed by the other two authors and discussed until agreement was reached on both raw data and higher order themes. These were then summarized in a way that would translate the participants’ in-depth discussions and feedback provided in an easily understood language [33].

## 3. Results

We had incomplete MTI data from three athletes, therefore ten athletes were included in the subsequent MTI analysis. The results demonstrated a significant main effect for MTI variability: *F*(2,26) = 31.4, η² = 0.707, *p* < 0.001. Bonferroni follow-up tests revealed significant differences between each level; baseline to minimum MTI score = −7.5 increments on MTI, (SE = 1.72, *t*(26) = 4.35, *p* ≤ 0.001, *d* = 1.14); baseline to maximum MTI score = 6.14 increments on MTI, (*SE* = 1.72, *t*(26) = −3.56, *p* = 0.004, *d* = −1.07); and minimum to maximum MTI scores = 13.64 increments on MTI, (*SE* = 1.72, *t*(26) = −7.91, *p* ≤ 0.001, *d* = −1.95). However, there was no effect of MT changing in a consistent pattern over the 30 days but rather appeared to be more related to particular antecedents on the day (we ran a one-way ANOVA with 5 levels (sessions 1,2,3,4,5) on pre-session MTI scores for the ten participants with complete data. The assumption of sphericity was not violated (W = 0.210, *p* > 0.05) and there was no significant differences in MTI score between sessions over time, *F*(4,40) = 0.823, *p* = 0.518. Therefore, the differences found between lowest and highest sessions were not due to a systematic effect over time). The MTI data are presented in Table 2. Coefficient of variation (CV) data for each participant from the four most pertinent daily survey items are presented in Table 3. CV demonstrated a range of 9–22% for sleep quantity, 11–29% for sleep quality, 20–74% for stress and 0–34% for significant other connection.

Coding of transcribed qualitative interviews that followed resulted in three higher-order themes, labeled as Thrive, Prepare and Activate with multiple secondary and tertiary themes identified, as shown in Figure 1. A table showing details on inclusion criteria, best quote, and related mental toughness research is available in Appendix A. 

### 3.1. Thrive

Thrive was identified to summarize the value placed by study participants on overall foundational well-being, which included both mind and body. The concept of thrive or thriving in this context is similar to that of being “engaged in person-context regulatory processes that eventuate in healthy and productive adult personhood” [34] (pp. 25) or a process of development that leads to attaining an ideal state of personhood [35] rather than that of a successful outcome [36]. Participants described the various elements going into this high-level theme as a form of cornerstone factor in their lives that, when present, would then allow them to store and then access a more considerable amount of mental toughness in pursuing a specific training session or event. For example, participant TS discussed the impact of general stress on overall mental toughness levels in saying:

“*External things—this was December, so I was thinking about races I have coming up in April, and all the planning is going on in December and thinking through that. I think it was wearing on me a little bit and probably some family stress coming into the holidays. Just what are we getting the kids and family stuff. Nothing revolutionary or ground-breaking or heart-breaking but just a different level of heightened anxiety through that month*”.

Interestingly, participants often referenced the elements under Thrive when there was a gap present. To “thrive” was to be in an optimal mental and physical state—the desired baseline from which they could then utilize mental toughness to move closer to their goal. The descriptions provided by participants typically referenced not necessarily a bonus, but rather something that was missing (from the desirable baseline) and would thus limit available levels of mental toughness, such as fuel or hydration noted here by RW:

“*If I am not adequately fueled—and that includes hydration—then I just can’t suffer. I know going into that workout that is an issue, and that is a tough one to overcome. There are certain things you can sometimes put behind you, but those are areas—my sleep and nutrition that are tough to overcome. If I am significantly dehydrated or significantly under-rested, I do not care. It is so much harder to push through that pain level. And in turn, going into it, it is harder to focus because it is always in the back of your head*”.

As noted above, the elite master runners were selected in part due to the inclusion of the realities of life in their experiences and responses. Rather than living in a relatively protective bubble (often experienced at least in part by professional athletes), they faced the pressures of careers, marriages, children, mortgage payments, and more in addition to the pressures related to their high-level athletic pursuits. MR commented about the impact of stress on the amount of mental toughness available with the following thoughts:

“*When I was in college, there was that type of stress. When I was married to my ex-wife, there was that stress. I am married now, and there’s mostly less stress. But yeah—we only have so much mental energy. If I go in at a much happier, stress-free baseline, then I think there’s more to tap into. I think that reservoir is far deeper than if you’re sort of running on an emotional drought. Then all of a sudden you just need something (and) then it is just not there*”.

There have been multiple references to several of the sub-themes noted here in prior mental toughness literature. Attitude [3], identifying as “tough” [37], values [38] and even self-efficacy [39]. However, they are generally seen in prior literature as characteristics demonstrated by those who already possess mental toughness rather than a driver of mental toughness variability and thus present the potential for a “chicken or egg” discussion about whether the cause of the variability or the presence of the mental toughness came first. While it is likely some combination of the two, the current study demonstrates the variability brought about by each theme appears to influence the level of mental toughness, as has been demonstrated in reference to the influence of sleep on mental toughness [40].

### 3.2. Prepare

The second of the high-level themes identified in this study was Prepare, based on the inclusion criteria of “make ready beforehand for some purpose, activity or use” [41]. This high-level theme included a variety of items identified by the participants that would benefit from advanced thought, planning and practice. The identification of a clear vision or specific goal was commonly mentioned and ranged from immediate (workout plan for that day) to longer-term projections, as described here by RW:

“*The other thing that was a limiter or caused me some issues is I just didn’t have that specific race or specific goal. I did not have something six weeks out of ‘this is what I am going to do. So, it was trying to maintain motivation when I did not have that set goal other than to be in shape*”.

Not only did the vision or goals need to be clear, but it was also vital that they were meaningful to the individual, as noted here by EK:

“*If my heart is in a goal that I have set for myself, then I can be very mentally tough. There have been times that I do not have a goal set in front of me or one that my heart is not in, and I find that my mental toughness decreases*”.

Mental toughness awareness, practicing and some form of “callusing” (or putting oneself in a situation that requires an increased level of mental toughness) was also part of the preparation aspects noted by the participants. GR points to the awareness piece:

“*By simply thinking about being “mentally tougher,” I think I was tougher during the hard efforts when I needed to be… Recording/reporting or at very least mentally checking in with myself before each harder effort day will be something I use to focus my efforts*”.(GR—questionnaire)

This was supplemented by feedback from NO, who noted the mental preparation component:

“*In training, I try to put myself in a tough position, specifically to create the opportunity to push through it. This gives me mental toughness for the actual event. I may not gain the desired result in training, but I can reflect to know what alteration needs to be made to achieve the success I want*”.(NO—questionnaire)

Some of the elements initially referenced under the theme of Thrive also demonstrated further benefits within the Prepare theme, in relation to task-specific application. For example, while athletes indicated the general (day by day) importance of nutrition, sleep, and hydration to building the foundation, they also pointed to the importance of specific planning leading up to the event or training session. GR (questionnaire) summarized such thoughts based on his own discoveries as a study participant, saying: “*I will also concentrate on sleep 3-5 days before my A-races, and will use caffeine on the morning of big efforts/races more regularly*”.

Similarly, while general self-efficacy [39] was noted as essential to overall thriving, task-specific self-efficacy and the necessity to integrate a focus on enhancing it in the moment played an essential role in the preparation phase, as RW indicated:

“*One of the big limiting factors is the thought ‘oh man—I do not know if I want to hurt that bad.’ It is eliminating those kinds of negative thoughts. Those thoughts defeat the purpose. Pain is part of the game, and sometimes I know I allow that to limit how far I push it. So, the approach that I would take would be just kind of saying ‘Hey—this is just fine.’ It is better than being in the dirt. Push as hard as you can because the pain makes you more alive*”.

A review of previous mental toughness literature provided similar results to what was revealed under the combined higher-level themes of Thrive and Prepare. The connection between mental toughness and the pursuit of or overcoming of a specific challenge tied to a goal pursuit is commonly referenced in the literature [42,43,44]. Similarly, as noted above with generalized self-efficacy [39], a connection between mental toughness and task-specific self-efficacy has been noted [45]. Also, studies on the value of strategic sleep, caffeine intake and other task-specific planning are common but none that relate these back to intra-individual variability in mental toughness. In contrast, studies tying together mental toughness with practice setting, coaching [46] and self-awareness [47] have been previously presented. In the case of self-awareness, the research was more generalized and not specific to awareness of mental toughness, but the conceptual overlap has been provided.

### 3.3. Activate

The final high order theme is Activate, which accurately summarized the tools and resources utilized by, or impacting, individuals during an activity to affect their ability to draw on available levels of mental toughness needed to achieve a specific outcome. Broadly, the primary sub-themes covered by Activate were attentional control, self-talk, and feedback. Attentional control [47] looks at the interaction between goal-directed (top-down) instructions and stimulus-driven (bottom-up) feedback. DM provided insight into this when sharing the following:

“*I think on the days where the voice is louder, and it is harder to silence, it is because there are so many other things going on in my life. You are more worn down whether it is physically worn down or mentally worn down going into a workout or a race. It gets harder and harder just to quiet the voice (saying) ‘I am going to push myself really hard.’ Because the brain is also going ‘well yeah—but you are also dealing with this and this and this and this and this.’ It is very easy when you’re just dealing with only the race, but when other things are going on around you, it is harder to quiet that voice that’s telling you to slow down*”.

Self-talk, which has previously been shown to influence appraisals and performance outcomes [48] was one of the most commonly referenced descriptors provided by the participating athletes. This self-talk generally took on two different forms: mantras and breaking down the task into pieces. JB spoke freely about the way she pulled a specific mantra from her previous battle with breast cancer that she now uses as a runner:

“*During my breast cancer thing, I had the phrase ‘Be brave. Be strong. Be badass.’ And it stuck with me, and when things get really hard in a race, I am just repeating that mantra again and again until things settle back in*”.

MR and others provided several examples of breaking the task into smaller pieces, of which this quotation provides a clear example:

“*If I am not sure, I can make it… then I will tell myself ‘go another 15 s.’ So rather than just saying ‘ok—it is over,’ I will say ‘no—just 15 s, and then I will reassess.’ And normally I can say ‘well that was not that bad… give me another 15…’ Once I get down to a minute then I start telling myself ‘I can do anything for a minute.’ So, what feels like forever—2:45—if I can chop it down and talk myself into just doing little sections and then get myself down to a minute*”.

Feedback also could have a potential positive or negative impact in the form of early performance (depending on how the “callusing” from the Prepare theme is utilized). TS provided a clear example of how this feedback (in this case mile time splits) may negatively influence levels of mental toughness with the following explanation:

“*If I am doing a tempo and I start at 6:20’s (pace/mile) and (plan to) go down to 5:30’s, but I cannot even get to 5:50, that starts to play in the mental game. ‘Oh, I should be running 20 s faster, but I cannot do that.’ So that is where the mental thing came in, and if it is not working for me, then I am throwing in the towel*”.

The projecting of “future self” into the self-talk discussion based on that early feedback was often described as lifting these high performing athletes to increased levels of mental toughness. This higher level was either because they were not willing to settle for what they might later consider being a mediocre performance or remembered how it would feel when they were done, as DM shares with the following:

“*Remember that feeling you had afterward—there’s no greater feeling than that. I think that is why we keep going back and keep pushing. We crave that kind of feeling. If I can hone in on that feeling before I go out the door for that workout—(knowing) I am going to feel so much better afterward. I think that is what helped pull me into mentally a good place to be able to get out the door and do it on the days when it is not ideal*”.

Attentional control has been identified as a critical mental toughness characteristic [49]. However, as with many of the elements discussed earlier, previous research on attentional control points to it being a characteristic of mental toughness rather than an influencer of that mental toughness variability, as identified here. Similarly, self-talk is noted as being something mentally tough people do more consistently than others [50] instead of the precursor to higher levels of mental toughness noted within this study. No references tying together mental toughness with thoughts of future self could be found. However, the feedback from others was seen to positively impact mental toughness in multiple studies [51,52].

## 4. Discussion

The purpose of this study was to explore whether mental toughness varies across a 30-day training block and whether such variability is associated with specific antecedents. As hypothesized, there were significant differences in mental toughness (based on MTI assessment) both from baseline and between high-RPE sessions (Table 2). Our results, therefore, supported the state [4] rather than trait [53] view of mental toughness. These differences were identified, despite the potential ceiling effects possibly caused by elite athletes reporting high scores on mental toughness inventories [54]. Additionally, this is unlikely to be due to significant inherent variability in the measurement tool, as prior research has supported the scale reliability—both between and within levels of analysis of the MTI across three different and independent samples [6]. Instead, the results support previous studies indicating at least a portion of mental toughness is state-like and can change over time [6,7]. The specific themes identified within our study also appear to support the hypothesized indicators of mental toughness identified by Gucciardi [6]. Self-efficacy was specifically identified in both studies and buoyancy (related to Activate), attention regulation (attentional control), emotional regulation, context and success mindset were indirectly referenced in our findings.

The exploratory element of the study sought to investigate potential reasons for this variability. Foundational wellbeing (Thrive) was the first of the higher order themes and incorporated all four of the items initially tracked within either the raw data or second order themes of the qualitative analysis. The second of the higher order themes (Prepare) included two of the four relevant items covered by the coefficient of variation analysis, including sleep and (indirectly) stress. The last of the three higher order themes (Activate) naturally did not include sleep since this theme involves what is happening in the moment. However, there was a clear connection between the raw data and the other two pertinent coefficients of variation items (Table 3). The importance of setting aside distractions (stress), focusing on the now (stress) and feedback from friends and family (significant other connection) were included in the raw data themes in this third area of emphasis.

We would, therefore, suggest a model describing mental toughness as an interaction between state and trait drivers, as described by Harmison [4] and Gucciardi et al. [6]. Capacity mental toughness may represent the maximum possible level of mental toughness a specific individual can attain (trait). Then, the concept of functional mental toughness [7] represents the actual mental toughness being accessed now (state) based on the application of the Thrive, Prepare and Activate components identified in this study. While an individual may have a high level of inherent, or capacity, mental toughness, that does not necessarily mean that the same individual is taking purposeful steps within the themes of Thrive, Prepare and Activate to optimize the amount of mental toughness that is being utilized functionally. 

Through the introduction of the terms Capacity Mental Toughness and Functional Mental Toughness, both researchers and practitioners may have an opportunity to assist individuals in optimizing their own performance and outcomes more effectively. These concepts help to build and enhance the practical application by athletes, researchers, and coaches that Gucciardi et al. [6] and Harmison [4] initiated. By outlining the trilogy of Thrive, Prepare and Activate, individuals, coaches, and practitioners will be able to more effectively contextualize specific action plans (e.g., practicing self-talk as part of the Prepare phase) as a specific step toward influencing the level of functional mental toughness for each. Strategies that would incorporate these specific steps provides for a much more productive discussion—and application potential—than merely comparing the mental toughness of person A to that of person B.

### 4.1. Application for Practitioners

One of the encouraging findings noted through this study was the duality of interaction both between and within each of the higher-level themes. Each of the themes (Thrive, Prepare, and Activate) positively influenced the level of available (functional) mental toughness, but that may not be the limit of the interaction. Instead, participants indicated that each one also has the potential to impact the others (i.e., increased preparation enhanced ability to effectively activate).

A further application element is that despite the positive between- and within-theme interaction, we also propose they stand alone regarding benefit. As such, the individual who is not thriving personally and did not adequately prepare mentally for the event or session can still benefit from utilizing tools/resources within the “activate” theme. While not receiving all available benefits, there are still potential increases in mental toughness garnered by engaging in each one separately. This potentially provides the individual, coach or peer advisor resources through which to enhance functional mental toughness regardless of current life or situation state on that day or in that specific situation.

The combination of these themes may provide practitioners a framework that can be utilized to help individuals identify critical areas for improvement and necessary steps to enhance outcomes through choices that best fit their current situation, history, and future goals. For example, a practitioner could assist a client in reflecting on their current level of foundational wellbeing (Thrive), their situation-specific advanced preparation (Prepare) or the tools utilized in the midst of the specific activity requiring the mental toughness (Activate). Such reflection could lead to the identification of specific personal development aspects that could be improved over time either independently or in tandem with a coach or practitioner. While not to be over-emphasized due to the inherent limitations herein, the practical application opportunities provided through these results are encouraging.

### 4.2. Limitations

While the small sample size adopted meant that our quantitative analysis (and hence generalizability) was limited, the qualitative approach adopted enabled us to explore mental toughness variability with each participant. The inclusion of non-professional and middle-aged athletes brings a specific “real life” element to this exploratory case study of individual mental toughness variability and potential influencers. These individuals are considering mental toughness and the elements driving variability of that mental toughness within conditions that are familiar to many athletes (e.g., family and work pressures). However, we acknowledge that self-regulation and psychological well-being generally increase with age [55] and that different themes may have been generated by younger athletes. Furthermore, by asking participants to consider mental toughness within the study, it may have drawn attention to issues not usually in the participants’ thoughts and potentially influenced the findings. We would argue that mental toughness is one of the most used construct for athletes and coaches [56]. However, the use of a naturalistic or deception approach might have limited this priming effect.

Additionally, the selected one-month period over which the data collection took place meant that the athletes were at different points in their seasons rather than all pursuing similar training outcomes. Some had recently come off of their crucial race periods, while others were just starting their build period for the coming year and this variation could have had an additional impact on outcomes. However, our focus on self-selected high-RPE sessions meant that the intensity of those sessions was left to the interpretation of each athlete and hence ensured individual differences in perceptions of ‘what makes a tough session’ could be identified. 

### 4.3. Future Directions

This study sets the stage for a range of future study directions. There is an opportunity to investigate each of the three primary optimizers: Thrive, Prepare and Activate, both individually and in combination. The interaction within each of these three influencers of mental toughness could also be examined more closely (e.g., whether preparation increases mental toughness which then increases the likelihood of preparing further). Researching the impact of the specific components under the thrive, prepare and activate influencers on mental toughness across a larger sample size is essential in order to provide greater certainty of application across the general population. This follow-up research could initially be done with a larger group of athletes and then be extended more broadly, outside of sport. The potential to investigate whether the functional enhancement of mental toughness can be applied outside of sport (i.e., military, executives, and well-being) would continue to expand the opportunity for such interventions to affect the lives of a broader population positively. The concept that foundational well-being positively influences mental toughness (and vice-versa) is an intriguing concept that could benefit society on a broader basis beyond athletics. For example, individuals could potentially put a higher priority on areas such as sleep, nutrition, and exercise if they realized these elements would benefit their level of mental toughness across other areas of their lives.

The practical implications introduced in this study provides additional traction for future research to investigate the concept of mental toughness influencers such as sleep and psychological training further. If mental toughness is merely a relatively constant trait, there is limited value tied to psychological skills training. However, the identification of variability and potential influencers means researchers, practitioners and individual athletes (and others) can integrate elements that can help improve their outcomes and their lives.

## 5. Conclusions

In conclusion, the findings from this study not only support previous research that mental toughness is state-like [6], but findings also suggest that there are many potential optimizers that can be utilized by individuals to improve their mental toughness and thus their outcomes. The current study highlighted specific influencers and helped categorize these influencers into an easy to understand trilogy of themes. This format may provide researchers, practitioners, and individuals additional tools and resources on which they can continue to build better outcomes and potentially better overall lives. There is still much to learn about how specific influencers can improve the utilization of mental toughness across a variety of settings, but this provides a starting point going forward.

## Figures and Tables

**Figure 1 sports-07-00156-f001:**
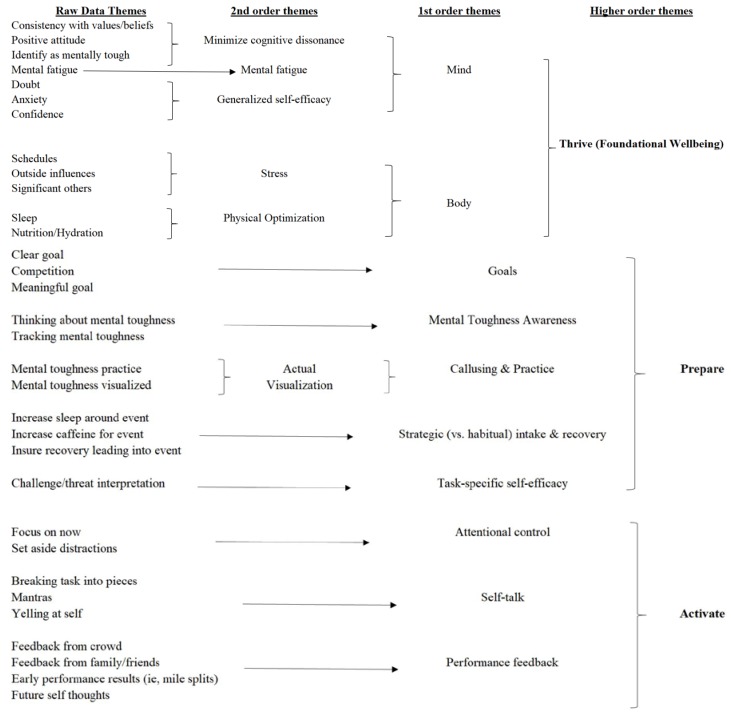
Themes uncovered from open coding of interview data.

**Table 1 sports-07-00156-t001:** Demographic information from the thirteen participants.

Pseudonym	Ranking Score	Age (years)	Gender (M/F)
SS	76.53	40	F
JB	76.25	41	F
WM	78.71	44	F
DM	85.3	43	F
EK	79	47	F
RS	91	59	M
PB	75.03	49	M
NO	75.5	48	M
TS	95.6	50	M
RW	85	52	M
TR	78.4	54	M
GR	77.8	46	M
MR	80	51	M
Average	81.1	48.0	
Standard Deviation	6.35	5.37	

Note: Ranking Score 100% = World record level; >90% = World class; >80% = National class; >70% = Regional class.

**Table 2 sports-07-00156-t002:** Mean (± SE) composite mental toughness index (MTI) score (range, 8–56) at baseline (outset of study) compared to lowest (minimum) and highest (maximum) reported scores during five sessions.

RM Factor 1	Mean	SE	95% Confidence Interval	
			Lower	Upper
Baseline	47.0	1.56	43.8	50.2
Minimum	39.5	1.56	36.3	42.7
Maximum	53.1	1.56	50.0	56.3

**Table 3 sports-07-00156-t003:** Overview of athlete survey data of those providing consistent responses. Each row represents a participant. SD = Standard Deviation; CV = Coefficient of Variation (SD/mean, expressed as a percentage).

Participant	Sleep Quantity			Sleep Quality			Stress			Significant Other		
	(In Hours)			(1–10 Scale)			(1–5 Scale)			(1–5 Scale)		
	Mean	SD	CV	Mean	SD	CV	Mean	SD	CV	Mean	SD	CV
DM	7.53	0.66	9%	6.87	1.52	22%	3.13	0.62	20%	4.81	0.40	8%
JB	6.12	1.11	18%	6.62	1.92	29%	2.15	0.92	43%	4.96	0.20	4%
MR	8.34	0.76	9%	6.10	1.47	24%	2.34	0.67	29%	3.93	0.53	13%
RW	7.42	0.93	13%	6.33	2.12	33%	2.87	1.38	48%	2.67	0.92	34%
TS	7.63	0.92	12%	8.22	0.93	11%	2.74	0.66	24%	2.44	0.51	21%
NO	7.14	1.57	22%	7.69	1.85	24%	1.52	0.63	41%	5.00	0.00	0%
PB	8.75	0.76	9%	8.03	1.05	13%	2.21	0.57	26%	4.90	0.54	11%
TR	7.34	0.73	10%	6.87	1.28	19%	2.39	0.67	28%	3.06	0.25	8%
GR	7.79	0.85	11%	8.25	1.26	15%	3.46	0.78	23%	3.50	0.72	21%
WM	7.81	0.69	9%	8.59	1.05	12%	1.48	1.09	74%	5.00	0.00	0%

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
