# Peer review of "An Exploratory Case Study of Mental Toughness Variability and Potential Influencers over 30 Days"

_sports, 2019, doi:10.3390/sports7070156_

Round 1
Reviewer 1 Report
Dear Authors,
Concerning the article, I have to point out the following positive factors:
- The topic address an importan theme for scientific community.
- The paper is well written and its interesting for the reader.
- The methodology is the correct and proper for this kind of studies and variables.
- The Vancouver format is well followed.
- The introduction help the reader understand the backgrounds of the scientific field that is treated in the whole study.
- Results express in many ways the response to the goal and the advance of the science that the paper gives.
As a whole, the paper deserves to be published in this journal, because it increases the advance of its scientific field. Regarding that point, I believe it should be published in this version without changes.
Yours faithfully,
Regards
Author Response
Thank you for your kind comments and encouraging review.
No further changes requested by this reviewer so no additional comments to be added here.
Reviewer 2 Report
Dear Authors,
Congratulations on tackling a very topical and perhaps timely issue, such as within-person mental toughness (MT) variability. The topic will be of great interest to the readership of Sports. Personally, I was interested in your research since I have collaborate with Dr. Gucciardi and I am using MTI extensively. I hope you find my comments useful.
There are not several concerns and areas where the study and manuscript requires strengthening.
The quality of the narrative is high. I did not struggle to read the manuscript in order to maximize my confidence in understanding the essence of what was done, why it was done, and what the findings mean for the field. The MS requires minimum grammatical adjustment and proofreading. The vast majority of the parts of the MS conform to APA standards.
Here are some suggestions of mine:
1. My main concern is about some conceptual issues. You state that Gucciardi conceptualizes MT as multi-dimensional, when he definitely does not support that idea. The multi-dimensionality is for example supported by MTQ48 and its developers. Yes, Daniel conceptualizes MT based on seven dimensions plus adversity (I would not call them "subordinate constructs") as it is evident in Table 2 in his 2015 paper where MTI was developed (reference #6). In addition, even in the 2017 paper that you also cite (#2), you can easily see that the "caravan or resources" he is talking about is about key dimensions that "...accumulate and integrate over time" (p. 96). So, I would suggest clarifying this crucial conceptual issue;
2. The Introduction and especially the first two paragraphs are not that intriguing. I see what connection you are trying to make, but it could be written in a better way. I would suggest you re-write at least the fist two paragraphs in a way that captures the reader's attention;
3. Was the period set to 30 days, because it just so happened based on the athletes' sessions or there was another reason?
4. Line 342: "We moved beyond..." How many raters were involved? Was there agreement? If not, how was it resolved? Please, give some more details about this part of the procedure; and
5. I would personally use Table 2 (mentioned above) and make some comments in the Discussion section about the themes that you found and the key dimensions Daniel is using to conceptualize MT. I see many similarities already. Also, in the right column of the same table, you can find many theories you could use in your Discussion. I think that would strengthen your Discussion and Suggestions substantially.
The Research Question is important and your work is original. It is also evident that a lot of hard work went into this paper. I hope that you will find my comments helpful.
Author Response
Thank you for taking the time to provide a valuable review of our manuscript. Point by point responses in the format requested by Sports has been provided in the attached document. We appreciate your feedback to help optimize the final version of this manuscript for publication.

Reviewer 3 Report
I would like to commend the authors on taking an innovate approach to investigating mental toughness in an athletic population. There are a number of interesting observations and I believe the manuscript will help develop critical thinking on this topic. The manuscript is well presented and provides good detail, with the inclusion of a number of direct quotations from the athletes a useful addition. However, at times I was left questioning if the findings were more related to personal motivation (and fluctuations in motivation) than specifically related to mental toughness. I do still believe the manuscript can make a useful addition to the current literature.
The following comments are made to assist the authors in developing the manuscript.
General comments:
Within the text (L.317-322) the suggestion is made that by participating in the research project and being aware that their MT was being assessed made the athlete more conscious of their MT. So the question is, has participation in the research project drawn attention to things not usually in the participants thoughts and potentially influenced the findings?
I believe that some further reference needs to be made to the age of the participants and the influence this may have. There is considerable research (from a range of settings) indicating that self-regulation and psychological well-being increase with age. With this knowledge I am wondering if these scores are influenced by a number of other factors and would the same variability be seen across a wider population.
Of interest is the athletes included in "group one" with higher variability in MTI scoring appear to athletes who are higher ranked (4 of the 5 top ranking are included in this group), with the highest ranking athlete (TS) having the highest variability. I would like to see some further consideration of this and the impact it may have on the findings.
Throughout the results and discussion more reference is needed to variability across time points. Much of the provided text states specific components of MT but it fails to clearly articulate how these have changed over the training period and how each athletes perceptions of MT has fluctuated. Can the authors articulate how MT varied over time and the affect of this on the athlete?
Specific comments:
L.47-49 - this direct definition has been used earlier in the text. Please revise
L.50-55 - the hypothesis and purpose needs to be moved to the end of the introductory text
L.56-57 - this sentence is awkwardly worded
Methods - a more comprehensive overview of the thematic analysis procedure is required. Please include the details of what happened at each step.
Methods - no reference is made to the development of trustworthiness. Can this be included.
L.176 - can examples of the questions and probes be provided
L.183 - there are a number of limitations to telephone based interviews. Can the authors acknowledge these.
Results / figure 1 - can the raw data themes (particularly in the 'Mind' 1st order theme) be split into facilitative or debilitative i.e. increased confidence and decreased anxiety. The point is that these raw data themes could have positive or negative effects
Results - the inclusion of quotations from the interview transcripts is highly beneficial, but can these be made clearer (i.e. put in italics and indented)
L.360-362 - this is a good example of variation that needs to be developed throughout the rest of the manuscript
L.381-387 - I am questioning if this is related to feedback or just a result of low mental toughness. Please expand on this point.
References - there are couple of errors within these. For example reference 7 has the title included twice
Author Response
Thank you for taking the time to provide a detailed review of our manuscript. We have provided responses in the format requested by Sports in the attached document.
Your recommendations will help optimize this manuscript for final publication and we are appreciative of the feedback you provided to help make that happen.
